# Efficient Deep Representation Learning by Adaptive Latent Space Sampling

## Abstract

Supervised deep learning requires a large amount of training samples with annotations (e.g. label class for classification task, pixel- or voxel-wised label map for segmentation tasks), which are expensive and time-consuming to obtain. During the training of a deep neural network, the annotated samples are fed into the network in a mini-batch way, where they are often regarded of equal importance. However, some of the samples may become less informative during training, as the magnitude of the gradient start to vanish for these samples. In the meantime, other samples of higher utility or hardness may be more demanded for the training process to proceed and require more exploitation. To address the challenges of expensive annotations and loss of sample informativeness, here we propose a novel training framework which adaptively selects informative samples that are fed to the training process. The adaptive selection or sampling is performed based on a hardness-aware strategy in the latent space constructed by a generative model. To evaluate the proposed training framework, we perform experiments on three different datasets, including MNIST and CIFAR-10 for image classification task and a medical image dataset IVUS for biophysical simulation task. On all three datasets, the proposed framework outperforms a random sampling method, which demonstrates the effectiveness of proposed framework.

## 1 Introduction

Recent advances in deep learning have been successful in delivering the state-of-the-art (SOTA) performance in a variety of areas including computer vision, nature language processing, etc. Not only do advanced network architecture designs and better optimization techniques contribute to the success, but the availability of large annotated datasets (e.g. ImageNet (Deng et al., 2009), MS COCO (Lin et al., 2014), Cityscapes (Cordts et al., 2016)) also plays an important role. However, it is never an easy task to curate such datasets. Collecting unlabeled data and the subsequent annotating process are both expensive and time-consuming. In particular, for some applications such as medical imaging, the annotation is limited by the available resources of expert analysts and data protection issues, which makes it even more challenging for curating large datasets. For example, it takes hours for an experienced radiologist to segment the brain tumors on medical images for even just one case.

On the contrary to supervised deep learning, human beings are capable of learning a new behaviour or concept through the most typical cases rather than accumulative learning for a lot of cases. Intuitively, we may ask: Is it really necessary to train a deep neural network with massive samples? Are we able to select a subset of most representative samples for network training which can save the annotation cost, improve data efficiency and lead to an at least equivalent or even better model? To the best of our knowledge, this is a less explored domain in deep learning and relevant applications, where a lot of efforts have been put into optimizing the network designs. Rather than improving the performance of a neural network given a curated training set, here we are more interested in how annotated samples can be more efficiently utilized to reach a level of performance. We consider such property as '*data efficiency*', namely how efficient a learning paradigm utilizes annotated samples to achieve a pre-defined performance measure.

In this paper, we propose a model state-aware framework for data-efficient deep representation learning, illustrated in Figure 1. The main idea is to mine 'harder' training samples progressively on the data manifold according to the current parameter state of a network until a certain criteria is fulfilled

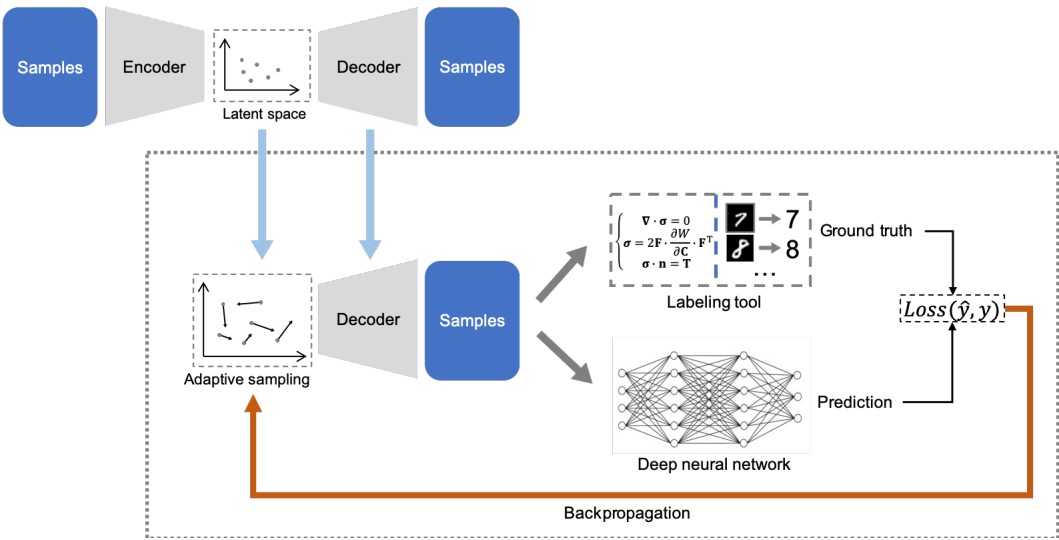

Figure 1: The general pipeline of proposed framework. The preparation stage is located at the top left corner which represents the training of a variational auto-encoder (VAE) using unannotated samples. The main stage is located within the dashed rectangle, where the decoder (generator) as well as its latent space are used for mining hard training samples according to the error information propagated backward via the target model and decoder (generator). Each proposed sample will be annotated by the labeling tool.

(e.g. size of training dataset or performance on validation dataset). The harder samples with respect to a given network state are defined as those yielding higher loss, which are estimated through back-propagation (Hinton et al., 2006). To be able to select plausible harder samples, a generative model is employed for embedding data into a low-dimensional latent space with better compactness and smoothness. In particular, we investigate two sampling strategies in the latent space, namely sampling by nearest neighbor (SNN) and sampling by interpolation (SI) for different applications. The data efficiency of our framework is evaluated on three datasets, including MNIST and CIFAR-10 for image classification tasks, as well as a medical image set IVUS for biophysical simulation task.

There are three major contributions of this work:

1. A general and novel framework is proposed for model state-aware sampling and data-efficient deep representation learning, which can be used in a variety of scenarios with high annotating cost.

2. Unlike previous studies (Sener & Savarese, 2017; Peiyun et al., 2019), a generative model is introduced to propose informative training samples. Two latent space sampling strategies are investigated and compared.

3. The framework is not only applicable for sampling on an existing dataset, but it also allows suggestive annotation and synthesizing new samples. We demonstrate the latter in a biophysical simulation task, where artificial samples are synthesized from the latent space.

## 2 RELATED WORK

### 2.1 FEW-SHOT LEARNING

In recent years, few-shot learning (FSL) has received a lot of attention, which is relevant to this work in terms of improving data efficiency in training. Few-shot learning was firstly proposed by Fei-Fei et al. (2006) and Fink (2005). Fei-Fei et al. (2006) proposed a Bayesian implementation which leverages knowledge from previously learned categories for one-shot image classification task, where some categories may contain only one training sample. Instead of directly learning for the few-shot tasks, similarity learning aims to learn the relevance of given two objects from a limited number of samples (Koch et al., 2015; Fink, 2005; Snell et al., 2017; Kulis et al., 2013; Vinyals

et al., 2016; Kim et al., 2019). For example, Koch et al. (2015) proposed the Siamese network for the few-shot classification by calculating the similarity between a labeled image and a target image. Snell et al. (2017) proposed a neural network that maps the samples into a metric space. Then the classification can be performed by computing distances to the prototype representations of each class. Vinyals et al. (2016) proposed to learn a deep neural network that maps a small labeled support set and an unseen example to its label, avoiding fine-tuning to adapt to new class types. It is noted that most FSL methods focus on the classification task. Some recent methods start to address other tasks such as object detection etc (Shrivastava et al., 2016; Schwartz et al., 2018; Zhang et al., 2019; Schonfeld et al., 2019). Schwartz et al. (2018) proposed to jointly learn an embedding space and the data distribution of given categories in a single training process which brought a new task into few-shot learning, namely few-shot object detection.

## 2.2 HARDNESS-AWARE LEARNING

This work is also related to the field of hardness-aware learning. The concept of mining hard examples has been employed in different machine learning paradigms for improving training efficiency and performance (Schroff et al., 2015; Huang et al., 2016; Yuan et al., 2017; Malisiewicz et al., 2011; Wang & Gupta, 2015; Zheng et al., 2019; Harwood et al., 2017). The main idea is to select those training samples that contribute the most to the training process. For example, Schroff et al. (2015) proposed the FaceNet which employed a triplet training approach. A triplet loss was used to minimize the distance between an anchor and a positive points and maximize the distance between the anchor and a negative points. The pairs of anchor and positive points are randomly selected, while hard pairs of anchor and negative points are selected according to a pre-set criterion. To address the issue of gradients being close to zero, Harwood et al. (2017) combined the triplet model and the embedding space by using smart sampling methods. The smart sampling method is able to identify those examples which produce larger gradients for the training process.

## 2.3 ACTIVE LEARNING

Our work is also highly related to active learning which has been widely studied (Settles, 2009). Sener & Savarese (2017) proposed to formulate active learning as a core-set selection problem. The core-set contains those points such that a model trained with them is competitive to a model trained with the rest points. Peiyun et al. (2019) were also addressing the similar problem as in this paper, investigating whether it is possible to actively select examples to annotate. They proposed an novel active learning paradigm that is able to assign a label to a sample by asking a series of hierarchical questions.

## 3 METHODOLOGY

Before we introduce the proposed framework in details, it is worth noting that all the quantities and notations used in the following sections are listed in Appendix A.1.

## 3.1 OVERVIEW

The proposed training framework consists of two stages as shown in Figure 1. The first stage can be considered as a preparation phase where a VAE-based generative model [1] is trained using unannotated samples. We obtain the generator (decoder) $G$ and the encoder $E$ with trained and fixed parameters as well as the $n$-dimensional latent space $\mathbb{R}^n$. In the second stage, 'hard' samples are mined iteratively in the latent space and fed into the network for training. The sampling strategies are based on the error information (normalized gradients) of the neural network model $F$ backpropagated through the generator $G$. In general, the hard samples can be either chosen from the candidates in the unannotated dataset or synthesized using the generator $G$. Accordingly, we investigate two sampling strategies named 'sampling by nearest neighbor (SNN)' and 'sampling by interpolation (SI)' as shown in Figure 2, which will be introduced in the following sections. The sampling and

---

[1] Here we use VAE as an example to demonstrate the idea. Other kinds of generative models can be also used in the framework accordingly.

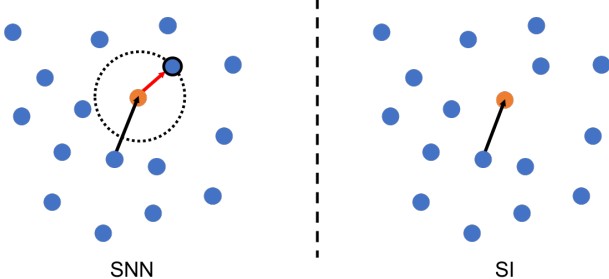

Figure 2: The figure demonstrates two sampling methods proposed in this paper: SNN (left) and SI (right).

training process are repeated until the stop criterion (the amount of training samples reaches a pre-set value) is satisfied.

In the following sections, the core processes of proposed training framework will be outlined. Extensive details to facilitate replication are given in Appendix A. As the training of a generative model is not a focus of this paper, details of the preparation phase are also given in Appendix A.

### 3.2 PROGRESSIVE TRAINING

As samples are mined and added to the train set progressively, we denote the original dataset of all samples as $\mathbb{D}$ and the mined set of hard samples used to training the network as $\mathbb{T}^{(t)}$, where $t$ denotes an index of round.

In the first round where $t = 1$, the first incremental subset $\mathbb{T}^{(1)}$ is constructed by randomly selecting a pre-set size of samples from $\mathbb{D}$ and sent to the labeling tools, resulting the initial train set $\mathbb{T}^{(1)} = \{(\boldsymbol{x}_1, \boldsymbol{y}_1), (\boldsymbol{x}_2, \boldsymbol{y}_2), ..., (\boldsymbol{x}_J, \boldsymbol{y}_J)\}^{(1)}$. For $t = 2, 3, ..., n$, the neural network model $F$ is first trained for a couple of epochs with the current train set $\bigcup_{k=1}^{t-1} \mathbb{T}^{(k)}$ using a given loss function $\mathcal{L}$. Then an incremental subset $\mathbb{T}^{(t)}$ with a pre-set size is constructed following the sampling strategies to propose more informative samples with respect to the current model state. The incremental subset $\mathbb{T}^{(t)}$ will be added to the train set $\bigcup_{k=1}^{t-1} \mathbb{T}^{(k)}$ forming $\bigcup_{k=1}^{t} \mathbb{T}^{(k)}$ for the next round of training.

### 3.3 ESTIMATING DIRECTIONS TOWARD HARDER SAMPLES

Our method is designed to mine the most informative samples for training the model. To achieve this, we randomly select a few annotated samples $\{(\boldsymbol{x}_1, \boldsymbol{y}_1), (\boldsymbol{x}_2, \boldsymbol{y}_2), ..., (\boldsymbol{x}_J, \boldsymbol{y}_J)\}$ from $\bigcup_{k=1}^{t-1} \mathbb{T}^{(k)}$ and identify the embedding position $\boldsymbol{p}$ for each $\boldsymbol{x}$ in latent spaces using $E$. Then, we evaluate the given loss function $\mathcal{L}$ with $\{(F(G(\boldsymbol{p}_1)), \boldsymbol{y}_1), (F(G(\boldsymbol{p}_2)), \boldsymbol{y}_2), ..., (F(G(\boldsymbol{p}_J)), \boldsymbol{y}_J)\}$ as:

$$\mathcal{L}_{\theta,\phi} = \Sigma_{k=1}^{J} L(F_\theta(G_\phi(\boldsymbol{p}_k)), \boldsymbol{y}_k) \tag{1}$$

where $F_\theta(G_\phi(\boldsymbol{p}_k))$ is the prediction via the generator $G_\phi(\cdot)$ and the target model $F_\theta(\cdot)$, and $L$ is any given metric. This would enable us to calculate the gradient $\frac{\partial \mathcal{L}}{\partial \boldsymbol{p}}$ for each sampling point $\boldsymbol{p}$ in the latent space $\mathbb{R}^n$ using the well-known back-propagation algorithm.

### 3.4 SAMPLING BY NEAREST NEIGHBOR

For each point $\boldsymbol{p}$ in latent space $\mathbb{R}^n$, now there is a corresponding gradient $\frac{\partial \mathcal{L}}{\partial \boldsymbol{p}}$ which is actually a direction to increase the loss $\mathcal{L}$. The normalized gradients is denoted as $\boldsymbol{d}$ for each $\boldsymbol{p}$. A new position $\boldsymbol{p}'$ that represents 'harder' sample for the model in the latent space $\mathbb{R}^n$ can be identified using Eq. 2

$$\hat{\boldsymbol{p}} = \boldsymbol{p} + \alpha \boldsymbol{d} \tag{2}$$

---

**Algorithm 1** The Proposed Framework Using Sampling by Nearest Neighbor

---

1: $t \leftarrow 1$
2: Randomly Initialise training subset $\mathbb{T}^{(1)} = \{(\boldsymbol{x}_1, \boldsymbol{y}_1), (\boldsymbol{x}_2, \boldsymbol{y}_2), ..., (\boldsymbol{x}_J, \boldsymbol{y}_J)\}^{(1)} \subseteq \mathbb{D}$
3: **while** Condition is not fulfilled **do**
4:     $t \leftarrow t + 1$
5:     Train $F$ with $\bigcup_{k=1}^{t-1} \mathbb{T}^{(k)}$ given loss function $\mathcal{L}$
6:     $\mathbb{S} \leftarrow \{(\boldsymbol{x}_1, \boldsymbol{y}_1), (\boldsymbol{x}_2, \boldsymbol{y}_2), ..., (\boldsymbol{x}_J, \boldsymbol{y}_J)\}^{(t)} \subseteq \bigcup_{k=1}^{t} \mathbb{T}^{(k)}$     ▷ Randomly select a subset
7:     $\boldsymbol{p} = E(\boldsymbol{x})$ for each $(\boldsymbol{x}, \boldsymbol{y}) \in \mathbb{S}$
8:     Evaluate $\mathcal{L}$ with $\{(G(\boldsymbol{p}_1), \boldsymbol{y}_1), (G(\boldsymbol{p}_2), \boldsymbol{y}_2), ..., (G(\boldsymbol{p}_J), \boldsymbol{y}_J)\}^{(t)}$
9:     Calculate $\frac{\partial \mathcal{L}}{\partial \boldsymbol{p}}$ for each $\boldsymbol{p}$ using backpropagation
10:     Identify harder point $\boldsymbol{p}'$ for each $\boldsymbol{p}$ by $\boldsymbol{p}' \leftarrow \boldsymbol{p} + \alpha \frac{\partial \mathcal{L}}{\partial \boldsymbol{p}}$
11:     Identify the nearest neighbor $\boldsymbol{p}$ for each $\boldsymbol{p}'$
12:     The new incremental set $\mathbb{T}^{(t)} \leftarrow \{(G(\boldsymbol{p}_1), \boldsymbol{y}_1), (G(\boldsymbol{p}_2), \boldsymbol{y}_2), ..., (G(\boldsymbol{p}_J), \boldsymbol{y}_J)\}^{(t)}$
13: **return** $F$

---

where $\alpha$ is a step size. For some scenarios where the labeling tool is not feasible to annotate a synthesized sample $G(\hat{\boldsymbol{p}})$, a nearest neighbor $\boldsymbol{p}$ from the embedding $E(\mathbb{D})$ will be found for $\hat{\boldsymbol{p}}$. Euclidean distance is used to identify the nearest neighbors.

Finally, the nearest neighbors can be used for retrieving their corresponding training pairs in $\mathbb{D}$ in order to construct a new incremental set $\mathbb{T}^{(t)} \leftarrow \{(G(\boldsymbol{p}_1), \boldsymbol{y}_1), (G(\boldsymbol{p}_2), \boldsymbol{y}_2), ..., (G(\boldsymbol{p}_J), \boldsymbol{y}_J)\}^{(t)}$ for the next round training. The pseudo code using sampling by nearest neighbor can be found in **Algorithm 1**.

### 3.5 SAMPLING BY INTERPOLATION

For scenarios where the labeling tool is capable to annotate arbitrary reasonable inputs (e.g. an equation solver), we propose an alternative sampling strategy called 'sampling by interpolation'. Synthesized samples are produced by the generator $G$ following the sampling strategy and used for training the neural network model.

At the first round $t = 1$, a set of variables $\{\boldsymbol{z}_1, \boldsymbol{z}_2, ..., \boldsymbol{z}_J\}^{(1)}$ in latent space $\mathbb{R}^n$ is drawn i.i.d from a multivariate normal Gaussian distribution. For each $\boldsymbol{z}$, the generator is adopted for generating the synthesized training samples which are annotated by the labeling tool $S$ obtaining the first incremental set $\mathbb{T}^{(1)} = \{(\boldsymbol{x}'_1, \boldsymbol{y}_1), (\boldsymbol{x}'_2, \boldsymbol{y}_2), ..., (\boldsymbol{x}'_J, \boldsymbol{y}_J)\}^{(1)}$. Here $\boldsymbol{x}'$ denotes the synthesized data generated by $G$ and $\boldsymbol{y}$ denotes the corresponding ground truth provided by $S$. Then, the first incremental set will be considered as the initial train set for training. For $t = 2, 3, ..., n$, the elements in $\mathbb{T}^{(t)}$ are the synthesized sample rather than existing samples. The incremental subset $\mathbb{T}^{(t)}$ will be added to the previous train set $\bigcup_{k=1}^{t-1} \mathbb{T}^{(k)}$ for training of next round.

Then, the process described in Section 3.3 would be followed such that the gradients $\frac{\partial \mathcal{L}}{\partial \boldsymbol{z}'}$ can be obtained from a loss function $\mathcal{L}$ using the back propagation. Once the gradients are derived for each point in latent space, we can use Eq. 2 again to identify a $\boldsymbol{z}'$ such that the generator $G$ can produce more informative sample $G(\boldsymbol{z}')$ for constructing a new incremental set. The pseudo code of proposed framework with sampling by interpolation is shown in **Algorithm 2**.

## 4 EXPERIMENTS

### 4.1 DATASETS

We evaluated the proposed framework on three different datasets, namely the handwritten digits classification — **MNIST** (Deng, 2012), image classification — **CIFAR-10** (Krizhevsky et al., 2009) and biophysical simulation — **IVUS** dataset. As MNIST and CIFAR-10 are well-known to the machine learning community, we will only introduce the IVUS dataset here. Intravascular ultrasound

---

**Algorithm 2** The Proposed Framework Using Sampling by Interpolation

---

1: $t \leftarrow 1$
2: Draw a set of latent space variables $\mathbb{Z}^{(1)} = \{z_1, z_2, ..., z_J\}^{(1)}$ i.i.d from a normal Gaussian.
3: Randomly Initialise training subset $\mathbb{T}^{(1)} = \{G(z_1), y_1), (G(z_2), y_2), ..., (G(z_J), y_J)\}^{(1)}$ where $y$ is provided by a labeling tool $\boldsymbol{y} = S(G(z))$
4: **while** Condition is not fulfilled **do**
5:     $t \leftarrow t + 1$
6:     Train $F$ with $\bigcup\limits_{k=1}^{t-1} \mathbb{T}^{(k)}$ given loss function $\mathcal{L}$
7:     $\mathbb{Z} \leftarrow \{z_1, z_2, ..., z_J\}^{(t)} \subseteq \bigcup\limits_{k=1}^{t} \mathbb{Z}^{(k)}$               ▷ Randomly select a subset
8:     Evaluate $\mathcal{L}$ with $\{(G(z_1), y_1), (G(z_2), y_2), ..., (G(z_J), y_J)\}^{(t)}$
9:     Calculate $\frac{\partial \mathcal{L}}{\partial z}$ for each $z$ using backpropagation
10:    Identify harder point $z'$ for each $z$ by $z' \leftarrow z + \alpha \frac{\partial \mathcal{L}}{\partial z}$
11:    The new incremental set $\mathbb{T}^{(t)} \leftarrow \{(G(z_1'), y_1), (G(z_2'), y_2), ..., (G(z_J'), y_J)\}^{(t)}$
12: **return** $F$

---

(IVUS) is a catheter-based medical imaging modality to identify the morphology and composition of atherosclerotic plaques of the blood vessels. The IVUS-derived structural parameters (e.g. plaque burden and minimum lumen area) are predictive in clinical diagnosis and prognosis. Imaged-based biophysical simulation of coronary plaques is used for assessing the structural stress within the vessel wall, which relies on time-consuming finite element analysis (FEA) (Teng et al., 2014). Here, we aim to train a deep neural network to approximate the FEA, which takes medical image as input and predicts a structural stress map. Our IVUS dataset consist of 1,200 slices of 2D gray-scale images of coronary plaques and corresponding vessel wall segmentation from the vendor. An in-house Python package named 'VasFEM' serves as a labeling tool by solving partial differential equations (PDEs) on segmentation masks. The details of relevant PDEs and FEA implementation are described in Appendix A.2.

## 4.2 EXPERIMENTAL PROTOCOL

The main purpose of experiments is to demonstrate the effectiveness of our proposed framework which can achieve the same performance using less training data. Thus, the performance of a given target model was evaluated using the same experimental setting (i.e. same optimizer, number of training iteration, batch size and size of training data).

The baseline method employed in this paper randomly selects a subset of training samples form real data until a certain condition is fulfilled, namely a predefined size of training samples. Instead of randomly choose training samplings, the proposed framework adopt aforementioned sampling strategies which progressively select or generate harder samples until a predefined condition is fulfilled.

An independent evaluation was carried out on the test sets of three dataset. For MNIST and CIFAR-10, the default test datasets which contain 10,000 testing samples respectively were used for evaluation the data-efficiency. For the IVUS dataset, a random held-out test dataset which contains 210 cases was evaluated using mean square error as performance metric. Two of them (MNIST, CIFAR) were performed with sampling by nearest neighbor since there is not a labeling system for the tasks, while experiments of IVUS was performed with sampling by interpolation where VasFEM was used for providing ground truth during training.

## 4.3 IMPLEMENTATION

For MNIST and IVUS datasets, a vanilla VAE (Appendix A.3) was trained using samples in train set without annotations. According to our preliminary experiments it is hard to reconstruct and generate the image for CIFAR-10 with a vanilla VAE. Thus, $\alpha$-GAN (Appendix A.3), proposed in Rosca et al. (2017), was employed which consists of four parts: 1) an encoder which map the image to the latent space, 2) a generator which reconstructs the image from the latent representation, 3) a code discriminator which can distinguish the learned latent distribution and the Gaussian distribution,

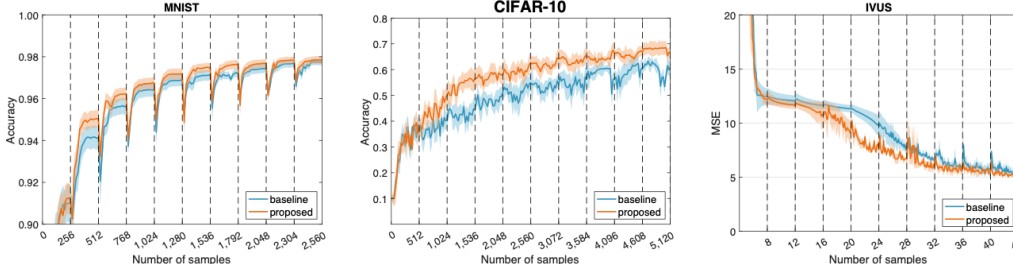

Figure 3: Comparison of the baseline method and the proposed method using different number of training samples on **MNIST**, **CIFAR-10** and **IVUS**

4) an image discriminator which can distinguish the reconstructed image and the real image. The dimensions of latent spaces for all three datasets are set to 3.

During training, convolutional neural networks (Appendix A.3) are employed for the classification tasks (MNIST and CIFAR-10) using the cross entropy loss. The biophysical simulation on the IVUS dataset is an image-to-image prediction task, for which deconvolution layers are employed in the target model and the mean square error was used as the loss function. Adam optimizer was used for all three tasks.

## 4.4 QUANTITATIVE RESULTS

For the experiment, we progressively increased the size of train set and reported the accuracy and mean square error (MSE) on the independent test sets respectively. Each experiment was repeated for five times for plotting the mean and variance (Figure 3). Not only a value of performance at each size (indexed by $t$) is reported (Table 1), the curves of accuracy and MSE are also included in Figure 3 after each increment of training samples. It is observed that the target model (denoted as 'Ours' in Table 1) trained under the proposed framework yields a better performance than the baseline (denoted as 'Rand' in Table 1) method until the target model reaches the bottleneck where the performance can hardly be improved by increasing size of the training samples. We also observed that the proposed framework yields a relatively low marginal performance improvement on MNIST and IVUS due to the simplicity of given tasks.

Table 1: A comparison of performance over different sizes of train set. For MNIST and CIFAR-10, higher accuracy represents better results. For IVUS, smaller MSE represents better results.

|  | $t$ | 1 | 2 | 3 | 4 | 5 | 6 | 7 | 8 | 9 | 10 |
|---|---|---|---|---|---|---|---|---|---|---|---|
| MNIST | Rand | 91.0 | 94.1 | 95.6 | 96.4 | 96.9 | 97.1 | 97.2 | 97.4 | 97.7 | 97.8 |
| Accuracy (%) | Ours | **91.2** | **95.1** | **96.2** | **96.7** | **97.2** | **97.5** | **97.6** | **97.7** | **97.8** | **97.8** |
| CIFAR-10 | Rand | 35.6 | 40.4 | 44.6 | 51.4 | 54.9 | 54.3 | 56.3 | 56.5 | 60.6 | 60.5 |
| Accuracy (%) | Ours | **37.0** | **50.0** | **56.7** | **58.0** | **62.7** | **64.2** | **65.3** | **66.5** | **66.9** | **66.9** |
| IVUS | Rand | 12.4 | 12.1 | 11.8 | 11.3 | 9.7 | 7.5 | 6.4 | 6.2 | 5.5 | 5.6 |
| MSE | Ours | **12.2** | **11.7** | **10.7** | **9.4** | **7.5** | **6.7** | **5.9** | **5.8** | **5.5** | **5.2** |

## 5 DISCUSSION

In our framework, an annotating system (i.e. labeling tool or original dataset) is integrated into the training process and used in an active manner. Based on the current model state, more informative samples proposed by a generator are annotated online and appended to the current train set for further training. This closed-loop design makes the most use of the annotating system, which would be very useful in scenarios with high annotation cost, e.g. medical image segmentation. From the performance curves in Figure 3, we observed an immediate drop when fresh samples were fed into the neural work. But the performance rebounded to a higher level as the neural network learned the information carried by these samples. Compared to the random sampling, our hardness-aware

sampling resulted in a deeper drop followed by a higher rebound, indicating that more informative sample were mined.

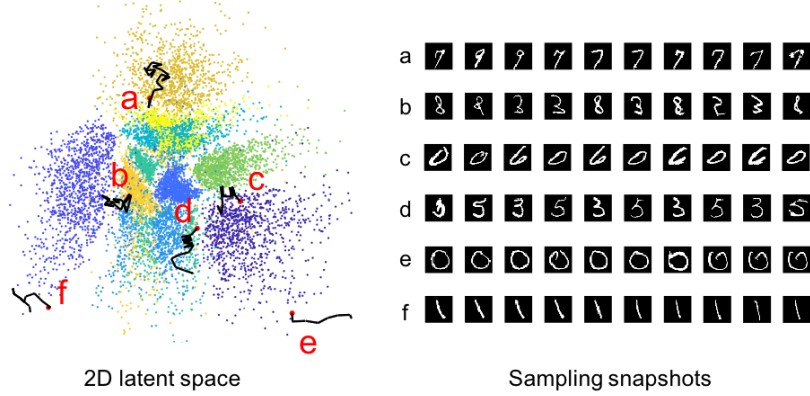

Figure 4: Latent space sampling on **MNIST**

The framework is able to mine new samples along the gradients of loss function. The samples mined in this way would bring more utility to training process than existing samples in the train set. To intuitively demonstrate this, a 2D latent space for MNIST embeddings is visualized on the left side in Figure 4. We choose a couple of initial sampling point and evolve them using rules of sampling by nearest neighbor without random re-selection, resulting in a collection of trajectories. Six typical trajectories are selected to be visualized in the 2D MNIST latent space. It can be observed that trajectories like **a**,**b**,**c** and **d** are the most desired exploration strategy in the latent space as they actually walk around the boundary between classes, where massive ambiguous samples are located. This provides us with a visual evidence that our framework indeed encourages to explore the areas of high uncertainty. The snapshots for every point in the corresponding trajectories are also visualized on the right side in Figure 4, which also provide strong support to our statement. There are two undesired trajectories denoted as **e** and **f**. We can see that **e** and **f** both have been initialized closed to outer boundary and they keep on exploring toward outside until there is no any nearest neighbor around. Such trajectories should be avoided by periodically re-selecting a new set of points within existing training samples for further exploring harder samples since they are not be able to provide informative samples any more.

Another crucial part is the choice of dimensionality of latent space. It is important for a generator to be capable to create plausible and diverse samples as they are in fact used for estimating more informative sample. Higher-dimensional latent space would encourage the diversity of synthesized samples, however, it would make the sampling points distributed in the latent space too sparse so that it will be difficult to identify the nearest neighbor for a given point in the latent space. Two sampling strategies along with the proposed framework were proposed for addressing different applications: 1) Sampling by nearest neighbor aims to handle the situation where there is not a external labeling tool and an existing annotated dataset is available. It is able to select the harder training samples from real dataset according to current training state. 2) Sampling by interpolation works when there is an external labeling tool (a FEM solver in our paper) available, which is able to provide ground truth for those synthesized training samples in a real-time manner. Synthesized training samples are more accurate in terms of the degree of difficulty since they are produced using the actual location in latent space instead of the nearest neighbors.

# 6    CONCLUSION

We proposed a model state-aware framework for efficient annotating and learning. Hard samples from the data manifold are mined progressively in the low-dimensional latent space. It is obvious that the proposed framework can not be only generalized to existing machine learning applications, but also those realistic scenarios as long as a labeling tool is available.

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

# A APPENDIX

## A.1 NOTATIONS

**Symbols and Notations**

| | |
|---|---|
| $\mathbb{D}$ | A The original training dataset |
| $\mathbb{T}$ | A incremental set that consist of training sample and its annotation |
| $\mathbb{S}$ | A temporary set which consist of randomly selected training samples. |
| $\mathbb{R}^n$ | A $n$-dimsional latent space |
| $\mathcal{L}$ | A given loss function |
| $G$ | The trained **generator** derived from a VAE-based model. |
| $E$ | The trained **encoder** derived from a VAE-based model. |
| $F$ | The target model that will be optimzed during training |
| $S$ | An external labeling tool providing synthesized training sample with a ground truth |
| $t$ | Round index |
| $\boldsymbol{x}$ | A training sample in its original space |
| $\boldsymbol{x}'$ | A synthesized training sample by generator |
| $\boldsymbol{p}$ | A point in the given latent space corresponds to a real training sample |
| $\boldsymbol{p}'$ | A point in the given latent space after displacement |
| $\boldsymbol{z}$ | A interpolated point in the given latent space |
| $\boldsymbol{z}'$ | A interpolated point in the given latent space after displacement |

## A.2 LABELING TOOL FOR IVUS DATASET

Imaged-based biomechanical analysis of coronay plaques were performed following the procedure described in (Teng et al., 2014). The workflow is wrapped into a Phython package named 'VasFEM' as a labeling tool, which is available upon request. The input to the labeling system is a segmentation mask of plaque and the output is the corresponding structural stress map with the same resolution.

The material of plaque is assumed to be incompressible and non-linear which is described by the modified Mooney-Rivlin strain energy density function:

$$W = c_1(\bar{I}_1 - 3) + D_1[e^{D_2(\bar{I}_1 - 3)} - 1] + \kappa(J - 1) \tag{3}$$

where $\bar{I}_1 = J^{-2/3}I_1$ with $I_1$ being the first invariant of the unimodular component of the left Cauchy-Green deformation tensor. $J = det(\mathbf{F})$ and $\mathbf{F}$ is the deformation gradient. $\kappa$ is the Lagrangian multiplier for the incompressibility. $c_1 = 0.138$ kPa, $D_1 = 3.833$ kPa and $D_2 = 18.803$ are material parameters of the blood vessels derived from previous experimental work (Teng et al., 2014). The finite element method is used to solve the governing equations of plane-strain problem:

$$\rho v_{i,tt} = \sigma_{ij,j} \qquad (i, j = 1, 2) \tag{4}$$

where $[v_i]$ and $[\sigma_{ij}]$ are the displacement vector and stress tensor, respectively, $\rho$ is the material density and $t$ stands for time.

A template pulsatile blood pressure waveform is applied on the lumen border. The structural stress map at the systole time point is extracted for analysis. It takes about ten mins to perform a 2D finite element analysis on a segmentation mask with 512x512 pixel IVUS image. As we focus on the data

efficiency of our proposed framework, we simplified the simulation by resampling the segmentation mask into 64x64 pixel image size and ignore the different components within the plaque. This reduced the simulation time to two mins. An example of the input image and output stress map is shown in Fig S1.

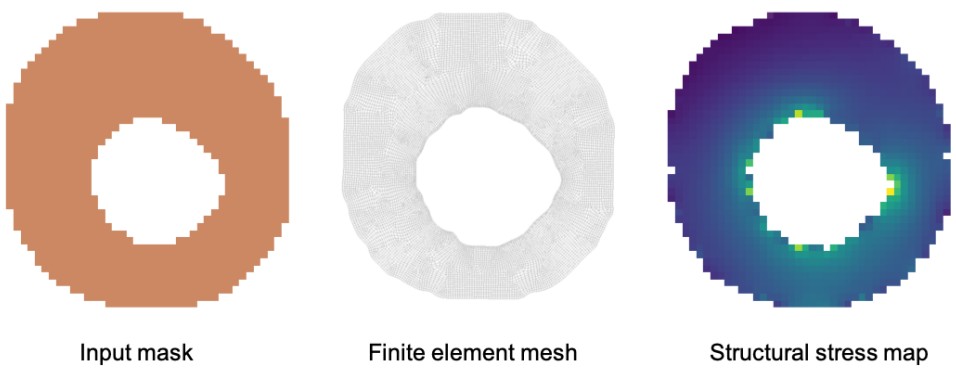

Input mask        Finite element mesh        Structural stress map

Figure S1: An example of the input and output of the labeling tool for IVUS dataset.

## A.3 STRUCTURE OF NETS

Table S1: The structure of the residual block for $\alpha$-GAN

| Operation | Kernel | Strides | Padding | IChannel | OChannel |
|---|---|---|---|---|---|
| 2D Convolution | 3 | 1 | 1 | IChannel | IChannel |
| Batch Normalization | - | - | - | - | - |
| Leaky-Relu Activation | - | - | - | - | - |
| 2D Convolution | 3 | 1 | 1 | IChannel | IChannel |
| Batch Normalization | - | - | - | - | - |
| Leaky-Relu Activation | - | - | - | - | - |

Table S2: The structure of the encoder for $\alpha$-GAN

| Operation | Kernel | Strides | Padding | IChannel | OChannel |
|---|---|---|---|---|---|
| 2D Convolution | 5 | 1 | 2 | 3 | 64 |
| Average Pooling | 2 | 2 | - | - | - |
| Relu Activation | - | - | - | - | - |
| Residual Block | - | - | - | 64 | 64 |
| Average Pooling | 2 | 2 | - | - | - |
| Residual Block | - | - | - | 64 | 64 |
| Average Pooling | 2 | 2 | - | - | - |
| Residual Block | - | - | - | 64 | 64 |

| Operation | Input | Output |
|---|---|---|
| Fully Connected Layer | 65536 | 3 |

Table S3: The structure of the generator for $\alpha$-GAN

| Operation | Input | Output | | | |
|---|---|---|---|---|---|
| Fully Connected Layer | 3 | 65536 | | | |
| Relu Activation | | | | | |
| **Operation** | **Kernel** | **Strides** | **Padding** | **IChannel** | **OChannel** |
| Residual Block | - | - | - | 64 | 64 |
| Average Pooling | 2 | 2 | | | |
| Residual Block | - | - | - | 64 | 64 |
| Average Pooling | 2 | 2 | | | |
| Residual Block | - | - | - | 64 | 64 |
| Average Pooling | 2 | 2 | | | |
| Residual Block | - | - | - | 64 | 64 |
| 2D Convolution | 1 | 1 | 0 | 64 | 3 |
| Tanh Activation | - | - | - | - | - |

Table S4: The structure of the image discriminator for $\alpha$-GAN

| Operation | Kernel | Strides | Padding | IChannel | OChannel |
|---|---|---|---|---|---|
| 2D Convolution | 5 | 1 | 2 | 3 | 64 |
| Average Pooling | 2 | 2 | | | |
| Relu Activation | | | | | |
| Residual Block | - | - | - | 64 | 64 |
| Average Pooling | 2 | 2 | - | - | - |
| Residual Block | - | - | - | 64 | 64 |
| Average Pooling | 2 | 2 | - | - | - |
| Residual Block | - | - | - | 64 | 64 |
| **Operation** | **Input** | **Output** | | | |
| Fully Connected Layer | 65536 | 1 | | | |

Table S5: The structure of the code discriminator for $\alpha$-GAN

| Operation | Input | Output |
|---|---|---|
| Fully Connected Layer | 3 | 700 |
| Leaky Rely Activation | - | - |
| Fully Connected Layer | 700 | 700 |
| Leaky Rely Activation | - | - |
| Fully Connected Layer | 700 | 1 |
| Sigmoid Activation | - | - |

Table S6: The structure of the target model for IVUS

| Operation | Kernel | Strides | Padding | IChannel | OChannel |
|---|---|---|---|---|---|
| 2D Convolution | 2 | 1 | 0 | 1 | 8 |
| Batch Normalization | - | - | - | - | - |
| Relu Activation | - | - | - | - | - |
| Max Pooling | 2 | 2 | - | - | - |
| 2D Convolution | 2 | 1 | 0 | 8 | 16 |
| Batch Normalization | - | - | - | - | - |
| Relu Activation | - | - | - | - | - |
| Max Pooling | 2 | 2 | - | - | - |
| 2D Convolution | 2 | 1 | 0 | 16 | 32 |
| Batch Normalization | - | - | - | - | - |
| Relu Activation | - | - | - | - | - |
| Max Pooling | 2 | 2 | - | - | - |
| 2D Convolution | 2 | 1 | 0 | 32 | 64 |
| Batch Normalization | - | - | - | - | - |
| Relu Activation | - | - | - | - | - |
| Max Pooling | 2 | 2 | - | - | - |
| 2D Convolution | 2 | 1 | 0 | 64 | 64 |
| Batch Normalization | - | - | - | - | - |
| Relu Activation | - | - | - | - | - |
| 2D Transpose Convolution | 2 | 1 | 0 | 64 | 64 |
| Batch Normalization | - | - | - | - | - |
| Relu Activation | - | - | - | - | - |
| Up-sampling | 2 | - | - | - | - |
| 2D Transpose Convolution | 2 | 1 | 0 | 64 | 32 |
| Batch Normalization | - | - | - | - | - |
| Relu Activation | - | - | - | - | - |
| Up-sampling | 2 | - | - | - | - |
| 2D Transpose Convolution | 2 | 1 | 0 | 32 | 16 |
| Batch Normalization | - | - | - | - | - |
| Relu Activation | - | - | - | - | - |
| Up-sampling | 2 | - | - | - | - |
| 2D Transpose Convolution | 2 | 1 | 0 | 16 | 8 |
| Batch Normalization | - | - | - | - | - |
| Relu Activation | - | - | - | - | - |
| Up-sampling | 2 | - | - | - | - |
| 2D Transpose Convolution | 2 | 1 | 0 | 8 | 1 |
| Sigmoid Activation | - | - | - | - | - |

Table S7: The structure of the target model for MNIST

| Operation | Kernel | Strides | Padding | IChannel | OChannel |
|---|---|---|---|---|---|
| 2D Convolution | 5 | 1 | 0 | 1 | 20 |
| Relu Activation | - | - | - | - | - |
| Max Pooling | 2 | 2 | - | - | - |
| 2D Convolution | 5 | 1 | 0 | 20 | 50 |
| Relu Activation | - | - | - | - | - |
| Max Pooling | 2 | 2 | - | - | - |
| **Operation** | **Input** | **Output** | | | |
| Fully Connected Layer | 800 | 500 | | | |
| Relu Activation | - | - | | | |
| Fully Connected Layer(Mean) | 500 | 10 | | | |

Table S8: The structure of the target model for CIFAR-10 (VGG 16)

| Operation | Kernel | Strides | Padding | IChannel | OChannel |
|---|---|---|---|---|---|
| 2D Convolution | 3 | 1 | 0 | 3 | 64 |
| Batch Normalization | - | - | - | - | - |
| Relu Activation | - | - | - | - | - |
| 2D Convolution | 3 | 1 | 0 | 64 | 64 |
| Batch Normalization | - | - | - | - | - |
| Relu Activation | - | - | - | - | - |
| Max Pooling | 2 | 2 | - | - | - |
| 2D Convolution | 3 | 1 | 0 | 128 | 128 |
| Batch Normalization | - | - | - | - | - |
| Relu Activation | - | - | - | - | - |
| 2D Convolution | 3 | 1 | 0 | 128 | 128 |
| Batch Normalization | - | - | - | - | - |
| Relu Activation | - | - | - | - | - |
| Max Pooling | 2 | 2 | - | - | - |
| 2D Convolution | 3 | 1 | 0 | 256 | 256 |
| Batch Normalization | - | - | - | - | - |
| Relu Activation | - | - | - | - | - |
| 2D Convolution | 3 | 1 | 0 | 256 | 256 |
| Batch Normalization | - | - | - | - | - |
| Relu Activation | - | - | - | - | - |
| 2D Convolution | 3 | 1 | 0 | 256 | 256 |
| Batch Normalization | - | - | - | - | - |
| Relu Activation | - | - | - | - | - |
| Max Pooling | 2 | 2 | - | - | - |
| 2D Convolution | 3 | 1 | 0 | 512 | 512 |
| Batch Normalization | - | - | - | - | - |
| Relu Activation | - | - | - | - | - |
| 2D Convolution | 3 | 1 | 0 | 512 | 512 |
| Batch Normalization | - | - | - | - | - |
| Relu Activation | - | - | - | - | - |
| 2D Convolution | 3 | 1 | 0 | 512 | 512 |
| Batch Normalization | - | - | - | - | - |
| Relu Activation | - | - | - | - | - |
| Max Pooling | 2 | 2 | - | - | - |
| 2D Convolution | 3 | 1 | 0 | 512 | 512 |
| Batch Normalization | - | - | - | - | - |
| Relu Activation | - | - | - | - | - |
| 2D Convolution | 3 | 1 | 0 | 512 | 512 |
| Batch Normalization | - | - | - | - | - |
| Relu Activation | - | - | - | - | - |
| 2D Convolution | 3 | 1 | 0 | 512 | 512 |
| Batch Normalization | - | - | - | - | - |
| Relu Activation | - | - | - | - | - |
| Max Pooling | 2 | 2 | - | - | - |

| Operation | Input | Output |
|---|---|---|
| Fully Connected Layer | 512 | 10 |

Table S9: The encoder of VAE for MNIST

| Operation | Input | Output |
|---|---|---|
| Fully Connected Layer | 784 | 400 |
| Relu Activation | - | - |
| Fully Connected Layer | 400 | 400 |
| Relu Activation | - | - |
| Fully Connected Layer(Mean) | 400 | 3 |
| Fully Connected Layer(Variance) | 400 | 3 |

Table S10: The decoder of VAE for MNIST

| Operation | Input | Output |
|---|---|---|
| Fully Connected Layer | 3 | 400 |
| Relu Activation | - | - |
| Fully Connected Layer | 400 | 400 |
| Relu Activation | - | - |
| Fully Connected Layer | 400 | 784 |

Table S11: The encoder of VAE for IVUS

| Operation | Input | Output |
|---|---|---|
| Fully Connected Layer | 4096 | 400 |
| Relu Activation | - | - |
| Fully Connected Layer | 400 | 400 |
| Relu Activation | - | - |
| Fully Connected Layer(Mean) | 400 | 3 |
| Fully Connected Layer(Variance) | 400 | 3 |

Table S12: The decoder of VAE for IVUS

| Operation | Input | Output |
|---|---|---|
| Fully Connected Layer | 3 | 400 |
| Relu Activation | - | - |
| Fully Connected Layer | 400 | 400 |
| Relu Activation | - | - |
| Fully Connected Layer | 400 | 4096 |

