# OpenReview forum: "Efficient Deep Representation Learning by Adaptive Latent Space Sampling"
_ICLR.cc/2020/Conference — Reject_

### Official Review · AnonReviewer1 · 2019-10-23
**Official Blind Review #1**

**Rating:** 3

**Review:**

Summary:
The paper proposes a method for sequential and adaptive selection of training examples to be
presented to the training algorithm. The selection happens in a latent space, based on choosing
samples which are in the direction of gradient of the loss in the latent space. Two selection
strategies are investigated: nearest neighbor and interpolation followed by generation. Results on
shown on MNIST, CIFAR10 and IVUS (Intravascular Ultrasound) datasets.


Detailed comments:
The proposed method works in two stages. First a VAE is trained using unannotated samples. In the
second stage, hard examples are found, in every iteration, in the latent space of the VAE and used
for sequential training. The sampling is done using the gradient of the objective function in the
latent space. The method makes sense, however the choice of space in which the sample selection is
being done is not well motivated or validated. The space could have been the original image space
(although given the high dimension, it would probably not work), or could have been any intermediate
feature space. Why was the space chosen to be the VAE latent space? Would it be possible to
demonstrate some benefits of doing so, theoretically and empirically?

The experiment section is relatively weak. The datasets used are relatively small and in two out of
the three datasets, the method does not improve. The dimension of the latent space is also
surprisingly small (2). While the main body of the paper describes the method with VAEs, the
experiments for the CIFAR10 dataset (where the results were in favor) were done \alpha-GAN.

**Experience Assessment:**

I have read many papers in this area.

**Review Assessment: Checking Correctness Of Derivations And Theory:**

I assessed the sensibility of the derivations and theory.

**Review Assessment: Checking Correctness Of Experiments:**

I carefully checked the experiments.

**Review Assessment: Thoroughness In Paper Reading:**

I read the paper at least twice and used my best judgement in assessing the paper.

---

> ### Author Response · Authors · 2019-11-12
> **Response to reviewer 1**
>
> We thank the reviewer for the constructive comments. We address the comments point-by-point below.
>
> 1. “Why was the space chosen to be the VAE latent space? Would it be possible to demonstrate some benefits of doing so, theoretically or empirically?”
>
> There are several reasons that motivate us to sample in the VAE latent space.
>
> Firstly, the latent space of a VAE model (or a deep generative model) plays a fundamental and important role in the proposed framework with the basic assumption that many high-dimensional data structures (e.g. images or patches) can be represented as a manifold in a low-dimensional space. Deep generative models enable us to identify such a manifold in the format of the latent code. Sampling in this latent space will generate plausible samples that follow the original data distribution. In contrast, sampling in the original data space or intermediate feature space might generate out-of-distribution samples. This is very important if we would like to use the ‘interpolation’ sampling method as discussed in Section 3.5.
>
> Secondly, the latent space also provides us with better representation and interpretation of data, for example, similar samples tend to be distributed nearby. As a result, the exploration trajectory becomes more explainable and meaningful, as discussed in Figure 4.
>
> Lastly, when we synthesize new samples, an interpolated point in the latent space can still produce a plausible training sample, compared to interpolating directly in the original data space. Therefore, when we navigate through the latent space according to the gradient direction, we can synthesize not only difficult but also plausible samples. The hardness information of synthesised training sample may be more accurate than the training sample selected by the nearest neighbours.
>
> 2. “The datasets used are relatively small and in two out of the three datasets, the method does not improve.”
>
> Although two of the datasets are relatively small, we still observed improvement according to the reported results (Figure 3 and Table 1). As Reviewer 3 pointed out, “The experiments are conducted on small-scale datasets like MNIST CIFAR-10, and IVUS MSE with satisfactory gain over the baselines.” More importantly, in this paper, we focus on proposing a novel framework and demonstrating its feasibility. Applying our framework onto large scale datasets will be a meaningful follow-on work for future research.
>
> 3. “The dimension of the latent space is also surprisingly small (2).”
>
> The latent spaces used for the three experiments (MNIST, CIFAR10, IVUS) have more than 2 dimensions. In the Appendix, the details of generative models and the corresponding latent space are given where you can find the actual numbers of dimensions (3 for all three datasets). We also discuss the choice of dimensions in Section 5.  In the revised version, we clarify the actual dimensions of latent spaces used for our experiments.
>
> In Figure 4, an example of 2-dimensional latent space was shown mainly for visualisation purposes and for discussing the exploration trajectories.
>
> 4. “While the main body of the paper describes the method with VAEs, the experiments for the CIFAR10 dataset (where the results were in favour) were done \alpha-GAN.”
>
> In this paper, we aim to demonstrate a general framework that adaptively selects more informative (harder) samples from the latent space to improve the training efficiency of a deep learning model. The deep generative model is a component of this framework. In the main body of the paper, we use the VAE as example to demonstrate the idea. But the framework can also use other kinds of generative models such as alpha-GAN etc as a component. We explicitly explained in Sec 4.3 Implementation that for the three tasks, two tasks (MNIST and IVUS) used VAE and other tasks (CIFAR-10) used alpha-GAN due to its better reconstruction performance on this dataset.
>
> More specifically, for the scenario where a labelling tool is not available, an auto-encoder-like generative model is essential for our framework since we need the correspondence between the latent space and original image space to identify the label of the selected training sample. For the scenario where a labelling tool is available, deep generative models like GAN can be used to generate informative training samples at the interpolated point in the latent space and the generated samples can be annotated subsequently by the labelling tool or human analyst.

---

### Official Review · AnonReviewer3 · 2019-10-23
**Official Blind Review #3**

**Rating:** 8

**Review:**

This paper proposes a method to efficiently select hard samples during the training of a neural network. This is achieved via a variational auto-encoder (VAE) that encodes the samples into a latent space. The VAE is trained in a preparation stage using the images only and fixed at later stage. During training of a DNN framework, samples are selected in the latent space and then decoded via the Decoder in VAE to generate the input for DNN framework. The advantage of such a framework is that now it is able to calculate the gradient w.r.t. the input samples of DNN. This gradient is used to determine the sampling strategy in the next iteration to select harder samples. Two different sampling methods are explored, including nearest neighbor and interpolation (with annotation tool step). The experiments are conducted on small-scale datasets like MNIST CIFAR-10, and IVUS MSE with satisfactory gain over the baselines. Overall, the paper is very well-written and easy to follow. Although the experiment results are not super exciting mainly because of small-scale datasets and not enough gain in the numbers, some of the analysis in Figure 4 are quite insightful to validate the assumption and motivation of this work. So I propose to accept this work for its novelty. I think this work will benefit future research in this direction.

However, I do have some concerns that I wish the authors could clarify if possible. First, the approach is very similar to online hard negative mining (OHNM) that is purely based on the loss to repeatedly select the samples that generate a larger loss. The major difference is that this work can model the sample distribution and thus select samples based on the gradient w.r.t. the samples in the latent space. This is very novel to me. However, I am wondering if the authors could compare with this sample baseline of OHNM. My concern is that the baselines in this work is too simple and it is not surprise that there is advantage over a simple baseline that is trained without any hard sample mining.

Second, the experiments are all conducted on small-scale and simple datasets like MNIST and CIFAR10. I am concerned how effective this approach could work for large-scale dataset. In the experiment, even for CIFAR10, a vanilla VAE will not work to reconstruct the input. So the authors have used alpha-GAN to help image reconstruction. If that is the case for CIFAR10 with only 10 classes, how could we extend this work to even larger dataset with more complicated background like ImageNet? I would think the preparation step itself is a very challenging task. This is my major concern that will question the effectiveness of the approach in real applications.

Third, a related question to the above one. As the input to the DNN is the reconstructed image from the pre-trained decoder, there will be some information loss during the reconstruction process. This is the major challenge, I think, for large-scale applications. Is that possible to use the original image as the input to DNN while still being able to find hard samples using the latent space and the image space correlation?

Fourth, I really like the visualization of Figure 4 that shows the trajectory of the sampling process that follows the boundaries between classes. The authors also mentioned that some trajectories explore towards outside util there is no real samples, which should be avoided. Could the authors comment on how to avoid such cases? In my understand, as the input is randomly sampled at the beginning, it cannot avoid such cases unless some evaluation is done during training to stop the sampling for these trajectories.

**Experience Assessment:**

I have published in this field for several years.

**Review Assessment: Checking Correctness Of Derivations And Theory:**

I carefully checked the derivations and theory.

**Review Assessment: Checking Correctness Of Experiments:**

I carefully checked the experiments.

**Review Assessment: Thoroughness In Paper Reading:**

I read the paper at least twice and used my best judgement in assessing the paper.

---

> ### Author Response · Authors · 2019-11-12
> **Response to Reviewer 3**
>
> We thank Reviewer 3 for appreciating the novelty of our work and the benefits for future research along the direction of data efficient learning. We address the comments point-by-point below.
>
> 1.Clarification of the difference from online hard negative mining (OHNM).
>
> In the field of hardness-aware learning, online hard negative mining (OHNM) or hard negative mining (HNM) selects the informative samples mainly by utilising the rank ordered by the training sample loss. This requires the whole dataset to be annotated before training. On the contrary, our framework is heuristic where the new informative training samples are identified within the neighbourhood of existing ones along the gradients derived from the loss function. We do not need to annotate the whole dataset and it is possible to perform human-in-the-loop learning. I.e., find the informative sample for annotation. This could dramatically benefit the tasks that have high cost on labelling each sample. We added this discussion to the Related Work Section.
>
> 2. The effectiveness of the method on large-scale and challenging datasets as the current work use 3 relatively small datasets.
>
> We agree with the reviewer that training a deep generative model for large-scale datasets and even “the preparation step itself is a very challenging task”. As the aim of this paper is to demonstrate the basic idea of the proposed training framework, we use the most popular datasets (MNIST and CIFAR10) in machine learning community as examples and an additional datasets (IVUS) to demonstrate the case when an online labelling tool is available. We also demonstrate that the proposed framework is flexible by using either VAE (for MNIST and CIFAR10) or alpha-GAN (for IVUS) as the generative model. For the large-scale cases where high-fidelity reconstruction is required, many recent SOTA works are based on either VAE or GAN [1-3]. We believe that the proposed framework is flexible to allow more advanced deep generative model to be integrated in future research.
>
> [1] Razavi, Ali, Aaron van den Oord, and Oriol Vinyals. "Generating Diverse High-Fidelity Images with VQ-VAE-2." arXiv preprint arXiv:1906.00446 (2019).
> [2] Gulrajani, Ishaan, et al. "Pixelvae: A latent variable model for natural images." arXiv preprint arXiv:1611.05013 (2016). (Results reported on ImageNet) [UPDATE: ICLR 2017]
> [3] Brock, Andrew, Jeff Donahue, and Karen Simonyan. "Large scale gan training for high fidelity natural image synthesis." arXiv preprint arXiv:1809.11096 (2018). [UPDATE: ICLR 2019]
>
> In addition, for cases where there is an available labelling tool, a GAN-based decoder is also acceptable to the framework, where harder samples can be synthesised to be annotated by the labelling tool in an online manner.
>
> 3. “As the input to the DNN is the reconstructed image from the pre-trained decoder, there will be some information loss during the reconstruction process. This is the major challenge, I think, for large-scale applications. Is that possible to use the original image as the input to DNN while still being able to find hard samples using the latent space and the image space correlation?"
>
> We understand your concern about the information loss during reconstruction. It is possible to replace the synthesised training samples with their corresponding original input. In fact, we noticed this issue when we were performing the experiments. On the other hand, because the datasets we used are relatively small-scale, currently we have not yet observed significant impact of the issue of information loss. Thus, to keep the description of methodology intuitive and simple, we did not include this alternative approach in the paper. But we believe that the alternative way would be a very promising extension for future work.
>
> 4. “I really like the visualization of Figure 4 that shows the trajectory of the sampling process that follows the boundaries between classes. The authors also mentioned that some trajectories explore towards outside util there is no real samples, which should be avoided. Could the authors comment on how to avoid such cases?"
>
> We really appreciate that you like Figure 4. To address the question, during the experiments, we periodically re-select a set of random points in the latent space as the new initial points for exploring harder samples, which would empirically reduce the chance of being trapped in the outer area. In the revised version, we further clarify this in Section 5.

---

> > ### Comment · AnonReviewer3 · 2019-11-12
> > **Thanks for the clarification**
> >
> > Thanks for the detailed comments. It has resolved my concerns. I think the paper is very interesting and insightful. We should encourage such work that explores how a method works. Although it is not practical for large-scale experiments yet, it may do with some extensions in future work. Therefore, I have raised my rating to "Accept" for this paper.

---

### Official Review · AnonReviewer2 · 2019-10-29
**Official Blind Review #2**

**Rating:** 6

**Review:**

This paper proposes a novel training framework which adaptively selects informative samples that are fed to
the training process. The adaptive selection or sampling is performed based on a hardness-aware strategy in the latent space constructed by a generative model.
The idea is intuitive and easy to follow. Experimental results demonstrate the efficacy of the proposed method.
I have two questions about this work:
1. The informative samples are fed to the training process, what about the rest "non-informative" ones?
2. What is the characteristics of the selected informative samples? i.e., for a class of images, which images should be informative?

**Experience Assessment:**

I have published in this field for several years.

**Review Assessment: Checking Correctness Of Derivations And Theory:**

I carefully checked the derivations and theory.

**Review Assessment: Checking Correctness Of Experiments:**

I carefully checked the experiments.

**Review Assessment: Thoroughness In Paper Reading:**

I read the paper thoroughly.

---

> ### Author Response · Authors · 2019-11-12
> **Response to reviewer 2**
>
> We thank Reviewer 2 for finding our work interesting and easy to follow. We answer the two questions below.
>
> 1.”The information samples are fed to the training process, what about the rest ‘Non-informative’ ones?”
>
> The proposed framework aims to improve the data-efficiency of a deep learning model. Although those relatively non-informative samples may be left out, the other more informative samples are already able to make the deep learning model achieve a desirable performance.
>
> Moreover, those relatively non-informative (left-out) samples would not be annotated, which significantly reduce the annotating cost.
>
> However, it should be noted that all samples including the non-informative ones are still necessary and useful to construct a compact latent space capturing the data distribution in the preparation stage.
>
> 2.”What is the characteristics of the selected informative samples? I.e., for a class of images, which images should be informative?”
>
> In our paper, Figure 4 illustrates 6 trajectories (4 of them are positive examples) of how informative training samples are selected. It shows that these trajectories are more likely to explore the boundary area between classes in the latent space where massive ambiguous training sample are located. For example, in Figure 4, trajectory “c” keeps sampling point between class “6” and “0” (also shown in the right panel as sampling snapshots) where they look quite similar to each other.  From both our visualisation and experiments, the characteristic of more informative samples is that they are more likely to be distributed around the boundary area between classes.

---

### Decision · Program_Chairs · 2019-12-19

**Decision:**

Reject

**Comment:**

VAE-based sample selection for training NNs.  A well-written experimental paper that is demonstrated through a number of experiments, all of which are minimal and from which generalization is not per se expected.  The absence of an underlying theory, and the absence of rigorous experimentation makes me request to extend either or, better, both.